# Effects of Various Additives on Fermentation, Aerobic Stability and Volatile Organic Compounds in Whole-Crop Rye Silage

**Horst Auerbach [1,*]** , **Peter Theobald [2]**, **Bärbel Kroschewski [3]** and **Kirsten Weiss [3]**

[1]   International Silage Consultancy, Thomas-Müntzer-Strasse 12, 06193 Wettin-Löbejün, Germany
[2]   Nürtingen-Geislingen University, Neckarsteige 6-10, 72622 Nürtingen, Germany; peter.theobald@hfwu.de
[3]   Faculty of Life Sciences, Albrecht Daniel Thaer-Institute of Agricultural and Horticultural Sciences, Humboldt Universität zu Berlin, Invalidenstrasse 42, 10115 Berlin, Germany; b.kroschewski@hu-berlin.de (B.K.); kirsten.weiss@hu-berlin.de (K.W.)
*   Correspondence: horst.uwe.auerbach@outlook.de; Tel.: +49-152-0168-3167

**Abstract:** Whole-crop cereal silage represents an important component of ruminant diets and is used as a substrate for biogas production. Due to the scarcity of data on whole-crop rye (*Secale cereale* L., WCR), our study aimed to evaluate the effects of a range of biological and chemical additives of different compositions on the fermentation and aerobic stability of silage made from this species. In addition, the production of various volatile organic compounds (VOCs), which potentially contribute to greenhouse gas emissions, was monitored. Regardless of additive treatment, all WCR silages were well fermented as reflected by the complete absence of butyric acid. Inoculants containing *Lactobacillus buchneri* and chemical additives reduced dry matter (DM) losses during fermentation for 53 days ($p < 0.001$), which were closely related with the concentration of ethanol upon silo opening ($R^2 = 0.88$, $p < 0.001$). Silage treated with *Lactobacillus buchneri*, alone or in combination with a homofermentative strain, had the lowest yeast count ($p < 0.001$) and, simultaneously, the highest aerobic stability ($p < 0.001$). Chemical additives outperformed all other additives by largely restricting the formation of ethyl esters of lactic and acetic acids ($p < 0.001$). The concentration of ethanol strongly correlated with those of ethyl lactate ($R^2 = 0.94$, $p < 0.001$), ethyl acetate ($R^2 = 0.85$, $p < 0.001$), and total ethyl esters ($R^2 = 0.94$, $p < 0.001$). The use of a simple linear regression model exclusively based on the ethanol content proved useful to predict the concentration of total ethyl esters in WCR silage ($R^2 = 0.93$, $p < 0.001$).

**Keywords:** aerobic stability; ethyl esters; fermentation; rye; silage additives; volatile organic compounds; VOC; whole-crop cereals; yeasts

## 1. Introduction

Whole-crop cereal silage made from wheat, barley, triticale, oats, and rye, has been widely produced as feed for ruminants [1], especially in climatic regions, which do not support the cultivation of maize [2]. In addition, it can be successfully used as a substrate for anaerobic digesters to produce biogas [3]. Dry matter (DM) yield, nutritive value and biogas production potential of the silage were shown to strongly depend on the stage of maturity at harvest, which is usually between early milk and dough stage [1,3–7].

To our knowledge, there is only one published study on the fermentation pattern of whole-crop rye (WCR) harvested at grain maturity, but data on DM losses during fermentation and aerobic stability (ASTA) were not reported [7]. On the contrary, numerous studies using barley, wheat, triticale and oats

at different stages of maturity have been performed showing poor fermentation quality as reflected by the presence of butyric acid [8,9], or rapid microbial deterioration upon exposure to air during feed-out [10–14]. The activity of clostridia and yeasts not only causes DM losses during fermentation, or feed-out, by producing $CO_2$ [15], but also decreases the nutritive value of the silage [16–18], thereby increasing silage production cost and reducing farm profitability.

The use of silage additives has been considered a flexible and a strategically promising management tool to alleviate the detrimental effects of undesired microbial activity, which contributes to minimizing losses in silage DM and nutritive value from field to trough. The types of additives—biological (homo- and heterofermentative lactic acid bacteria alone or in combination) and chemical additives (e.g., formic, propionic, sorbic and benzoic acids and their salts alone or in combination, sodium nitrite and hexamine, alone or in combination)—and their potential effects have been extensively reviewed [18–23]. However, these reviews were mainly conducted as meta-analyses, and therefore, lacked the specificity regarding crop type and silage additive composition. In addition, whole-crop cereal (WCC) silage was greatly underrepresented in the data sets used in comparison to maize, grass and lucerne silage. More so, trials on the effects of additives in WCC silage mainly tested biological products and with the exception of the studies by Nadeau [1] and by Knicky [9], no trial included both additive types.

Agriculture in general and forage production in particular have been put under scrutiny regarding their impact on the environment, including greenhouse gas emissions [24–28]. The review by Hafner et al. [29] focused on the range of volatile organic compounds (VOCs) produced in and potentially emitted from silage, their sources and environmental implications. Weiss [30], who suggested esters be considered as indicator VOC substances, highlighted the role of management factors affecting the formation of VOCs with particular emphasis on volatile organic acids, alcohols, and esters. Although there is a large body of evidence showing the potential of, in particular, chemical additives (composed of sodium benzoate, potassium sorbate used alone or mixtures thereof) to reduce VOC concentrations, the studies almost exclusively used maize silage [31–34], grass silage [30], and sugarcane silage [35,36]. Only Gomes et al. [13] tested the effects of a heterofermentative inoculant in whole-crop cereal silage made from oats.

Due to a general lack of data on WCR silage, our study aimed to evaluate the effects of a range of biological and chemical silage additives of different compositions on fermentation pattern, aerobic stability and VOC formation. Our hypothesis was that additives will affect the fermentation characteristics and aerobic stability (ASTA) of WCR silage according to their specific mode of action, known mainly from other silage types, and that the production and composition of VOCs can be altered by additive use.

## 2. Materials and Methods

### 2.1. Ensiling

Winter rye (*Secale cereale* L., *cv.* Protector, Saatenunion, Isernhagen, Germany) was grown on a dairy farm in Klein Schulzendorf, Germany (52.1949552 N–13.250923 W). Before seeding on 11 September 2016 at $3 \times 10^6$ seeds $ha^{-1}$, cattle slurry was applied as a sole N fertilizer source at a rate of 60 kg N $ha^{-1}$. In early March 2017, the field received additional N fertilizer (80 kg N $ha^{-1}$) from a mix of ammonium nitrate and calcium carbonate. The forage was directly cut with no wilting at the milk stage on 15 June 2017, and chopped by a precision chopper (Big X 650, Maschinenfabrik Bernard Krone GmbH & Co. KG, Spelle, Germany) set at a 30 mm theoretical particle length. The material was collected from different field areas, thoroughly mixed and composited to minimize the effect of sampling location on silage variables due to different forage composition [37]. The composition of the crop prior to additive application is given in Table 1.

**Table 1.** Composition of whole-crop rye before the additive application (*n* = 3, data given in g kg$^{-1}$ dry matter (DM) unless stated otherwise).

| Item | Mean | Standard Deviation |
|:---:|:---:|:---:|
| Dry matter, g/kg | 439 | 3.4 |
| Crude ash | 33 | 1.3 |
| Crude protein | 44 | 0.9 |
| Crude fibre | 362 | 2.0 |
| Starch | 79 | 4.5 |
| Sugar * | 157 | 3.9 |
| Water-soluble carbohydrates † | 214 | 4.0 |
| Metabolizable energy, MJ kg$^{-1}$ DM | 10.2 | 0 |
| Net energy lactation, MJ NEL kg$^{-1}$ DM | 6.1 | 0 |
| Lactic acid bacteria, log$_{10}$ cfu g$^{-1}$ | 5.4 | 0.14 |
| Yeasts, log$_{10}$ cfu g$^{-1}$ | 5.3 | 0.17 |
| Moulds, log$_{10}$ cfu g$^{-1}$ | 5.1 | 0.45 |

* sum of glucose, fructose and saccharose; † sum of sugar and fructosans.

The total amount of forage was divided into six piles, each of which was assigned to one treatment. After the manual additive application by spraying on the quantity of forage required for one replicate silo (1000 g), the material was packed into 1.5 L glass jars (Weck, Öfingen, Germany). The following additives (provided by KONSIL Europe GmbH, Wettin-Löbejün, Germany) were tested: CON, no additive (tap water); LAB$_{ho}$, homofermentative LAB, composed of *Lactobacillus plantarum* DSM 16,627 and *Lactobacillus paracasei* NCIMB 30,151 (total inoculation rate: 1.5 × 10$^5$ cfu g$^{-1}$), LAB$_{heho}$, combination of hetero- and homofermentative LAB, composed of *Lactobacillus buchneri* CNCM-I 4323 and *Pediococcus acidilactici* DSM 11,673 (total inoculation rate: 1.67 × 10$^5$ cfu g$^{-1}$); LAB$_{he}$, heterofermentative LAB solely composed of *Lactobacillus buchneri* CNCM-I 4323 (inoculation rate: 1 × 10$^5$ cfu g$^{-1}$); NHS, aqueous mixture of sodium nitrite (195 g L$^{-1}$), hexamethylene tetramine (71 g L$^{-1}$), potassium sorbate (106 g L$^{-1}$), applied at 2 mL kg$^{-1}$; BSP, aqueous mixture containing sodium benzoate (257 g L$^{-1}$), potassium sorbate (154 g L$^{-1}$) and ammonium propionate (57 g L$^{-1}$), added at 1.5 mL kg$^{-1}$. Biological additives were suspended in tap water so that the intended inoculation rate was achieved by applying 10 mL kg$^{-1}$ forage, whereas chemical additives were diluted with tap water to 10 mL kg$^{-1}$ forage.

The jars were equipped with a hole (diameter: 6 mm) in the lid and in the body, which were closed by rubber stoppers. The low DM packing density of 154 kg m$^{-3}$ was used to ensure rapid and free air penetration and circulation in the silage mass [38] to promote fungal development after rubber stoppers were removed for 24 h on days 28 and 46 of storage. Triplicate silages per treatment were produced to give a total of 18 silos, which were stored for 53 days in a dark room, whose temperature was set at 21 °C.

*2.2. Dry Matter Determination*

Forage and silage DM content was determined by oven-drying at 60 °C until constant weight was achieved, followed by 3 h of drying at 105 °C. Silage DM content was corrected for the loss of volatiles during drying [39], and the losses of DM during fermentation were calculated using the equation by Weissbach [40].

*2.3. Chemical and Microbiological Analysis*

Prior to chemical analysis, the samples were stored at −18°C. The German official methods for feed evaluation were employed to determine nutrient composition [41] and energy concentration [42] in oven-dried (60 °C, 24 h) forage samples.

Silage extracts were prepared by blending 50 g of silage with 200 mL of distilled water to which 1 mL toluene was added. After storage overnight at 4 °C, the extracts were filtered through a paper filter, followed by microfiltration (0.45 µm). Silage pH was measured potentiometrically by using a pH

electrode, and a colorimetric method based on the Berthelot-reaction (CFA, Scan++, Skalar Analytical, Breda, The Netherlands) was used to determine the ammonia-N concentration. Volatile fatty acids, alcohols and esters were analysed by GC with flame-ionization detection according to Weiss et al. [32], whereas lactic acid was determined by HPLC coupled with a refractive index (RI) detector [43].

The concentration of water-soluble carbohydrates (WSCs) in the forage was determined on oven-dried samples (60 °C, 24 h) after cold water extraction for one hour, whereas the extract for the determination of organic acids and alcohols was used for analysis in fresh silage samples. After the addition of the anthrone reactant composed of anthrone, thiourea and sulphuric acid, a spectro-photometrical analysis was performed, which mainly detects glucose, fructose, disaccharides, and fructosans [44].

For microbiological analyses, serial 10-fold dilutions of forage and silage samples were prepared in 0.1% (w/w) peptone water broth. Lactic acid bacteria were counted after pour-plating and incubation for four days on MRS De Man, Rogosa, Sharpe (MRS) medium (Oxoid, Basingstoke, UK) at 30 °C [45]. Yeasts and moulds were enumerated after spread-plating on yeast extract-dextrose-chloramphenicol agar and incubation for three to five days at 25 °C [46].

### 2.4. Aerobic Stability Test

The ASTA of silage was evaluated based on the temperature development in the samples compared with that of the room [47], which was kept at 20.6 ± 0.2 °C. Data loggers (Tinytag Talk 2, Gemini, Chichester, UK) were put in the geometric centre of a plastic container, which was loosely filled with silage. The temperature of the silage and of the room was recorded at 2 h intervals. Each plastic container was stored in an insulating polystyrene box allowing free air circulation for 336 h. Aerobic stability was defined as the number of hours before the temperature of the silage mass increased by 2 °C above room temperature.

### 2.5. Statistical Analysis

All data were analysed as a completely randomized design using SAS 9.4 software (SAS Institute Inc., Cary, NC, USA). Treatments were compared in a framework of a fixed effects model using silage additive as an experimental factor with three replications: $y_{ij} = \mu + \alpha_i + e_{ij}$, where $y_{ij}$ is the observed value of the jth replication from silage additive i; $\mu$ the population mean; $\alpha_i$ the fixed effect of silage additive i; and $e_{ij}$ the random residual effect of the ith treatment and jth observation, $\sim N(0, \sigma^2_e)$.

Microbiological data were $\log_{10}$-transformed prior to the statistical analysis. When fungal numbers were below the detection limit of 100 cfu $g^{-1}$ (= $\log_{10}$ 2.0), then the value was set at half the detection limit of 50 cfu $g^{-1}$, or $\log_{10}$ 1.7 cfu $g^{-1}$. After checking the assumption of normally distributed residuals by the Shapiro–Wilk Test and graphical diagnostics, traits for which the normality of observations could be assumed were subjected to ANOVA considering variance homogeneity, except for ethanol concentration. A non-parametric rank procedure using ANOVA-type statistics was employed for the non-normally distributed data (SAS procedure MIXED). When significance was detected in the global *F* test at $p < 0.05$, pairwise comparisons among means were performed by Tukey's test or by pairwise rank tests. Differences among means were reported as significant when $p < 0.05$, and trends were declared at $0.05 \leq p < 0.10$. In addition, the *t*-test was used to assess the selected contrasts between treatments containing *Lactobacillus buchneri* ($LAB_{he}$ and $LAB_{heho}$) and chemical additives (NHS and BSP).

The SAS procedure REG was employed to characterise the relationships between silage traits. The best fitted regression model (linear or various quasilinear) was selected at $p < 0.05$ (*F* test) based on the root mean square error (RMSE) and the coefficients of determination ($R^2$), which were adjusted for degrees of freedom.

## 3. Results

### 3.1. Dry Matter Losses, Fermentation Pattern, Aerobic Stability and Formation of Volatile Organic Compounds

The efficiency of the fermentation process was improved by reducing the DM losses in the treatments $LAB_{he}$, $LAB_{heho}$, NHS and BSP, but only chemical additives restricted WSC utilization during the fermentation process (Table 2). Based on the ammonia-N concentration, silage inoculated with $LAB_{he}$ and $LAB_{heho}$ showed the highest level of proteolysis. The additive NHS decreased lactic acid production, which was paralleled by the highest pH. Acetic acid formation was stimulated by both additives containing *Lactobacillus buchneri*. The use of $LAB_{ho}$ tended to decrease the acetic acid content when compared with untreated silage ($p = 0.090$). Other short-chain fatty acids, including propionic and butyric acids, were not detected in any of the silages.

**Table 2.** Effects of the additive type and composition on dry matter (DM) losses, fermentation characteristics, fungal populations and ester concentrations of whole-crop rye silage stored for 53 days (data presented as LS means in g $kg^{-1}$ DM unless stated otherwise, $n = 3$).

| Item | CON | $LAB_{ho}$ | $LAB_{he}$ | $LAB_{heho}$ | NHS | BSP | SEM | $p$ |
|---|---|---|---|---|---|---|---|---|
| DM, g $kg^{-1}$ | 419 [ab] | 413 [a] | 426 [bc] | 428 [c] | 429 [c] | 432 [c] | 1.7 | <0.001 |
| DM loss, % | 7.2 [b] | 8.2 [b] | 4.9 [a] | 5.2 [a] | 3.9 [a] | 4.1 [a] | 0.29 | <0.001 |
| Water-soluble carbohydrates | 62.5 [a] | 55.4 [a] | 50.7 [a] | 40.2 [a] | 131.1 [b] | 138.2 [b] | 6.23 | <0.001 |
| $NH_3$-N, g $kg^{-1}$ total N | 13.5 [bc] | 12.6 [b] | 13.9 [c] | 14.0 [c] | 11.2 [a] | 12.7 [b] | 0.24 | <0.001 |
| pH | 3.94 [bc] | 3.98 [cd] | 3.78 [a] | 3.81 [a] | 4.00 [d] | 3.92 [b] | 0.010 | <0.001 |
| Lactic acid | 33.8 [b] | 33.8 [b] | 33.3 [b] | 32.6 [b] | 27.8 [a] | 31.5 [b] | 0.70 | <0.001 |
| Acetic acid | 8.9 [a] | 5.8 [a] | 19.6 [c] | 21.0 [c] | 13.3 [b] | 12.7 [b] | 0.73 | <0.001 |
| Propionic acid | ND | ND | ND | ND | ND | ND | - | - |
| Butyric acid | ND | ND | ND | ND | ND | ND | - | - |
| Ethanol | 27.1 [c] | 28.9 [c] | 5.6 [ab] | 7.4 [b] | 1.5 [a] | 4.1 [ab] | 1.10–4.50 | 0.002 |
| n-propanol | ND | ND | ND | ND | ND | ND | - | - |
| 1,2-propanediol | 0.4 [x] | 0 [w] | 1.1 [y] | 1.4 [z] | 0 [w] | 0 [w] | 0–0.01 | <0.001 |
| Yeast count, $log_{10}$ cfu $g^{-1}$ | 6.6 [b] | 7.1 [b] | 3.5 [a] | 2.8 [a] | 6.7 [b] | 6.3 [b] | 0.26 | <0.001 |
| Mould count, $log_{10}$ cfu $g^{-1}$ | ND | ND | ND | ND | ND | ND | - | - |
| Aerobic stability, hours | 19 [w] | 15 [w] | 336 [y] | 303 [y] | 52 [x] | 63 [x] | 0–32.7 | <0.001 |
| Propyl acetate, mg $kg^{-1}$ DM | ND | ND | ND | ND | ND | ND | - | - |
| Ethyl lactate, mg $kg^{-1}$ DM | 307 [z] | 359 [z] | 127 [xy] | 127 [y] | 11 [w] | 62 [x] | 5.0–60.5 | <0.001 |
| Ethyl acetate, mg $kg^{-1}$ DM | 108 [z] | 87 [yz] | 44 [xy] | 35 [x] | 0 [w] | 0 [w] | 0–20.7 | <0.001 |
| Total ethyl esters, mg $kg^{-1}$ DM | 415 [z] | 446 [z] | 171 [xyz] | 162 [y] | 11 [w] | 62 [x] | 5.0–81.0 | <0.001 |

CON, no additive; $LAB_{ho}$, homofermentative inoculant composed of *L. plantarum* DSM 16627 and *L. paracasei* NCIMB 30151, total inoculation rate: $1.5 \times 10^5$ cfu $g^{-1}$; $LAB_{he}$, inoculant solely composed of *L. buchneri* CNCM-I 4323, inoculation rate: $1 \times 10^5$ cfu $g^{-1}$; $LAB_{heho}$, inoculant composed of *L. buchneri* CNCM-I 4323 and *P. acidilactici* DSM 11673, total inoculation rate: $1.67 \times 10^5$ cfu $g^{-1}$; NHS, aqueous chemical mixture containing sodium nitrite (195 g $L^{-1}$), hexamethylene tetramine (71 g $L^{-1}$) and potassium sorbate (106 g $L^{-1}$), 2 mL $kg^{-1}$; BSP, aqueous chemical mixture containing sodium benzoate (257 g $L^{-1}$), potassium sorbate (154 g $L^{-1}$), sodium propionate (57 g $L^{-1}$), 1.5 mL $kg^{-1}$; ND, not detected; [a–d] means in rows having no superscript in common differ at $p < 0.05$, Tukey's test; [w–z] means in rows having no superscript in common differ at $p < 0.05$, non-parametric rank test of ANOVA-type statistics, corresponding to the SEM in case of non-normally distributed data was calculated separately for each treatment.

The largest concentration of ethanol was found in untreated and $LAB_{ho}$-inoculated silage. Production of 1,2-propanediol was suppressed to below the detection limit in silage treated with $LAB_{ho}$, NHS and BSP, and its highest concentrations were detected in silage inoculated with $LAB_{he}$ and $LAB_{heho}$. No silage contained n-propanol.

Upon silo opening, large differences between treatments were observed in the yeast count and the aerobic stability, whereas moulds were not present in any of the silage produced. Inoculants containing *Lactobacillus buchneri* had the lowest yeast numbers and, simultaneously, showed the highest aerobic stability. Chemical additives improved ASTA over that of untreated and $LAB_{ho}$-inoculated silage by an average of 40 h ($p < 0.05$).

Propyl acetate was not detected. The chemicals NHS and BSP outperformed all other treatments in terms of restricting the formation of ethyl lactate, with a tendency found for BSP silage to contain a

lower concentration of this ester than was detected in $LAB_{he}$-treated silage ($p = 0.069$). Ethyl acetate synthesis was completely inhibited by the additives NHS and BSP, and a trend was observed for $LAB_{he}$ to reduce its concentration in comparison with $LAB_{ho}$ ($p = 0.063$). The total ethyl ester concentration was lowest in the treatments NHS and BSP. There were tendencies reported for significant differences in the total ester content for the comparisons between untreated silage and $LAB_{he}$ ($p = 0.052$), between $LAB_{ho}$ and $LAB_{he}$ ($p = 0.055$), and between $LAB_{he}$ and BSP ($p = 0.068$).

### 3.2. Relationships between Silage Traits

A strong positive polynomial relationship was detected between the concentration of ethanol at silo opening and the DM losses during fermentation (Figure 1, $R^2 = 0.88$, RMSE = 0.58, $p < 0.001$). Acetic acid content was very closely related with the yeast count ($R^2 = 0.86$, RMSE = 0.67, $p < 0.001$) and the ASTA ($R^2 = 0.91$, RMSE = 0.38, $p < 0.001$) (Figure 2).

Yeast count (x) and ASTA (y) were inversely related and their relationship was best described by a linear function (y = 542.5 − 74.53x, $R^2 = 0.92$, RMSE = 40.5, $p < 0.001$). In addition, a weak relationship was detected between the WSC concentration (x) at the silo opening and the ASTA (y) (y = 407.34 − 141.61xln(x), $R^2 = 0.22$, RMSE = 123.4, $p < 0.05$).

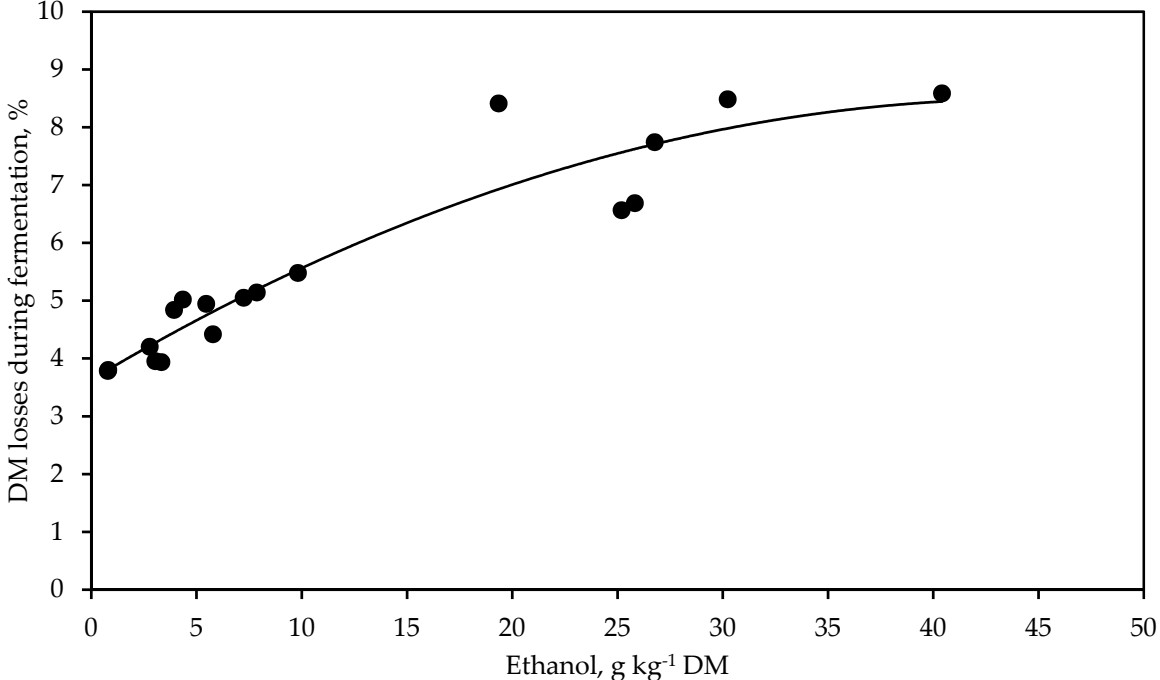

**Figure 1.** Relationship between the concentration of ethanol and the dry matter (DM) losses during fermentation in whole-crop rye silage treated with different additives and stored for 53 days. Y = 3.63 + 0.217x − 0.0024x$^2$, $R^2 = 0.88$, RMSE = 0.58, $p < 0.001$, $n = 18$.

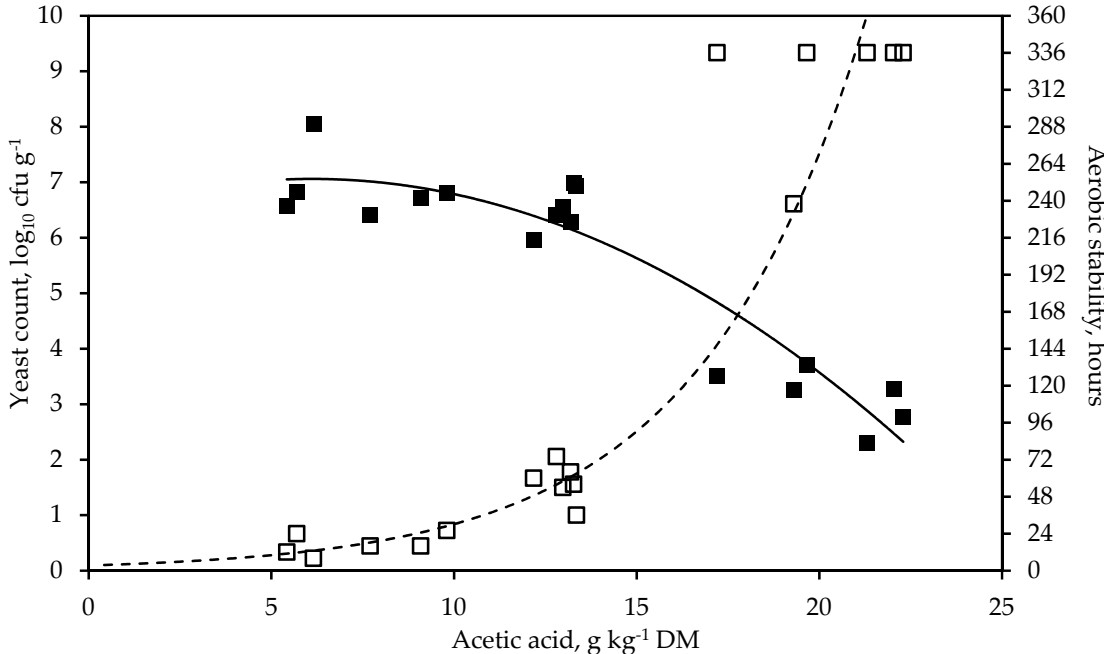

**Figure 2.** Relationship between the concentration of acetic acid and the yeast count (solid line, y = 6.39 + 0.221x − 0.018x$^2$, R$^2$ = 0.86, RMSE = 0.67, *p* < 0.001, *n* = 18) and the aerobic stability (dashed line, y = 3.37xe$^{0.219x}$, R$^2$ = 0.91, RMSE = 0.38, *p* < 0.001, *n* = 18), respectively, in whole-crop rye silage treated with different additives and stored for 53 days.

Very strong linear relationships existed between the concentrations of ethanol and and those of ethyl lactate (R$^2$ = 0.94, RMSE = 33.21, *p* < 0.001), ethyl acetate (R$^2$ = 0.85, RMSE = 17.67, *p* < 0.001), and total ethyl esters (R$^2$ = 0.94, RMSE = 45.84, *p* < 0.001), respectively (Figure 3).

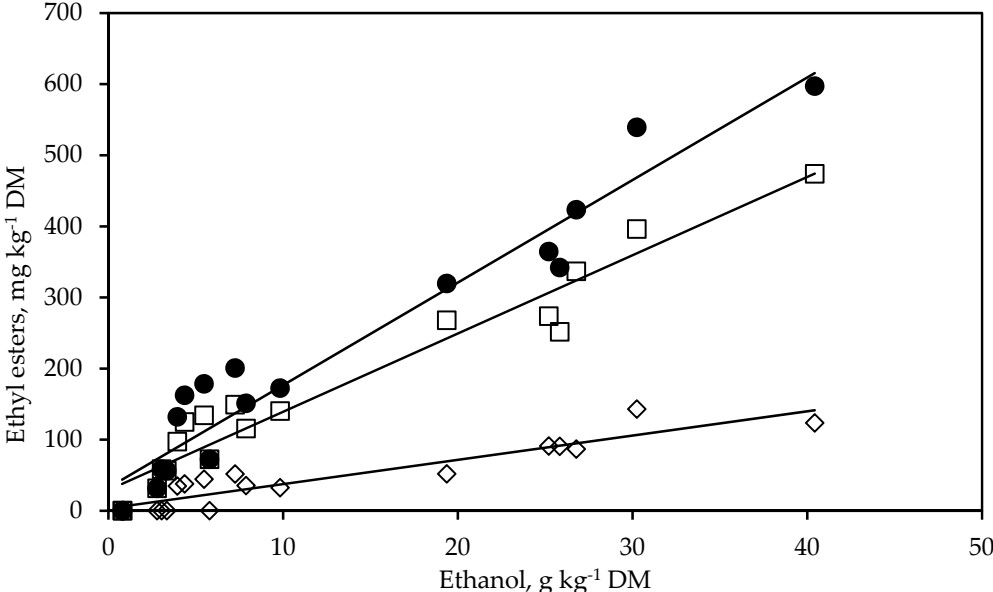

**Figure 3.** Relationships between the concentration of ethanol and the concentrations of ethyl lactate (□, y = 11.01x + 29.10, R$^2$ = 0.94, RMSE = 33.21, *p* < 0.001, *n* = 18), ethyl acetate (◊, y = 3.38x + 3.41, R$^2$ = 0.85, RMSE = 17.67, *p* < 0.001, *n* = 18), and total ethyl esters (●, y = 14.42x + 32.48, R$^2$ = 0.94, RMSE = 45.84, *p* < 0.001, *n* = 18) in whole-crop rye silage treated with different additives and stored for 53 days.

## 4. Discussion

### 4.1. Dry Matter Losses and Fermentation Characteristics

According to Borreani et al. [17], in-silo losses during fermentation can be as low as 5% under good silo management conditions on commercial farms. However, our results concerning untreated WCR silage supported previous observations of DM losses higher than that in laboratory ensiling experiments using various whole-crop cereal species [1,9]. With the exception of $LAB_{ho}$ inoculation, all additives were successful in our study to decrease DM losses to at least 5.2%, or lower, which is very close to the value suggested by Borreani et al. [17]. Regarding the effect of chemical additives on silage DM losses, our data confirm previous findings but contradict those for $LAB_{ho}$, which were also shown to have a positive effect. However, the magnitude of the effect was frequently found to be larger by chemical than by biological additives [1,9,19]. Data presented on DM losses disagree with observations by Kleinschmit and Kung [21], who found lower DM recovery, or higher DM losses, in their meta-analysis on the sole use of *Lactobacillus buchneri* in grass and small grain silage, by Filya [11] in whole-crop wheat, and by Gomes et al. [13] in whole-crop oats. However, others detected no effect of dual-purpose inoculants containing *Lactobacillus buchneri* and homofermentative species on DM losses in whole-crop barley silage [12,48], whereas Romero et al. [49] reported reduced losses in whole-crop oat silage, substantiating our results on WCR silage. The observed differences between studies regarding the additive effect on DM losses may be explained by differences in DM affecting the fermentation intensity [13], by different storage length and conditions (strict anaerobic storage vs. storage with repeated air ingress), and by variation in the epiphytic bacterial and fungal microbiome between cereal species [50], and even between varieties within species [51]. It is worthwhile mentioning here that, based on the assessment of a selected contrast ($p < 0.01$) comparing the *Lactobacillus buchneri*-containing inoculants $LAB_{he}$ and $LAB_{heho}$ (mean DM loss: 5.1%) with the chemical additives NHS and BSP (mean DM loss: 4.0%), our data confirm previous observations on a consistently larger magnitude of the effect by chemical additives [18,19].

Ultimately, DM loss during fermentation reflects the effect of additives on the fermentation pattern. Regardless of additive treatment, lactic acid concentrations in our study were within the typical range suggested for the silage of this DM content [52], and the significantly lower content in treatment NHS can be considered of no biological relevance. The lack of effect of $LAB_{ho}$ inoculation to stimulate lactic acid was likely caused by a relatively high number of epiphytic LAB on the crop at ensiling ($2.6 \times 10^5$ cfu $g^{-1}$) so that the added concentration of bacteria ($1.5 \times 10^5$ cfu $g^{-1}$) was not sufficient to dominate the fermentation process, probably due to the presence of highly competitive epiphytic species. On the contrary, in other studies on whole-crop barley or whole-crop rye cut before ear emergence, an inoculation rate of $1 \times 10^5$ cfu $g^{-1}$ (applied by additives composed of different homofermentative strains) proved successful to outcompete the epiphytic LAB flora [12,53]. This highlights the prominent role of the epiphytic bacteria flora and the strains used in homofermentative inoculants regarding the potential to affect lactic acid production. This is also substantiated by the lack of an effect of $LAB_{ho}$ treatment to reduce acetic acid production, which has frequently been observed [22,53,54]. When compared with untreated silage, *Lactobacillus buchneri* application usually causes lower lactic acid concentrations by the anaerobic degradation of this organic acid to form acetic acid, 1,2-propanediol, and minor quantities of ethanol [55], but this was not found in our study. Therefore, we assume that this unique metabolic pathway was not facilitated to a large extent, which is supported by the presence of only minute concentrations of 1,2-propandiol in silage inoculated with this species, and in untreated silage. Thus, it is more likely that the higher acetic acid production originated directly from the utilisation of pentoses by the phosphogluconate/phosphoketolase pathway [15], and/or by altered glucose metabolism induced by air infiltration in the silage, yielding acetate instead of ethanol. These oxygen-dependent changes in sugar metabolism have been described by Condon [56] for the obligately heterofermentative *Leuconostoc* species, but it remains unclear as to whether *Lactobacillus buchneri* also has this capacity. The aforementioned metabolic pathways would also explain why the

ethanol concentration in our study was lower than in untreated silage although an increase in ethanol content has often been associated with *Lactobacillus buchneri* use [13,21,57]. The effects of chemical additives, including those containing sodium nitrite, hexamine, sodium benzoate and/or potassium sorbate, on the production of organic acids and ethanol, substantiate the existing body of evidence from previous findings on whole-crop barley and wheat, grass and early-cut rye [9,18,57], of which the pronounced restriction of ethanol formation has been most prominent. The frequently observed larger ethanol-reducing effect of chemical additives over *Lactobacillus buchneri*-containing inoculants in various silage types [18,57] was confirmed by evaluating the contrast (*t*-test, $p < 0.01$) between the treatments $LAB_{he}/LAB_{heho}$ (mean content: 6.4 g $kg^{-1}$ DM) and the chemical additives NHS/BSP (mean content: 2.8 g $kg^{-1}$ DM).

The fermentation products n-propanol and propionic acid were not detected in our study. These are assumed to mainly originate from the activity of *Lactobacillus diolivorans* utilising 1,2-propanediol under anoxic conditions [58]. The most plausible explanation for this observation is that the epiphytic bacterial population did not contain this species. Had this species been present on the forage, its poor osmotolerance may have inhibited its ability to grow at higher DM concentrations [13,18]. Moreover, the very low concentration of the substrate for this fermentation pathway—1,2-propanediol—was insufficient to support the production of these two metabolic end-products. In addition, the fermentation length of 53 days may have been too short for n-propanol and propionic acid to occur. This assumption is substantiated by findings on high-moisture corn by da Silva et al. [59] showing that propionic acid and n-propanol levels increased with progressing storage length.

## 4.2. Aerobic Stability

Our data are in line with observations that silage may rapidly deteriorate upon air exposure [18], and that the use of homofermentative LAB was no successful strategy to improve aerobic stability [11,19,22]. They further substantiate the large existing body of evidence of positive effects of *Lactobacillus buchneri*, applied alone or in combination with homofermentative LAB, on the aerobic stability of different silage types, by inhibiting yeasts [12,13,18,21,23,48,49,54,57]. However, the lack of effect on yeast count and the weak improvement of ASTA by treatment with the tested chemical additives containing antifungal substances, were inconsistent with previous observations on grass, legume, maize and high-moisture corn silage [18,34,60–64]. Obviously, as the additive application rate was shown to affect the magnitude of the effect on ASTA [63], the dosage tested in our study may have been too low to exert a more pronounced improvement in ASTA. Auerbach and Nadeau [65] tested the same additive BSP in a series of maize silage trials of very similar experimental design to our WCR study and showed that, under challenging conditions of repeated air ingress during storage and short storage length, a higher application rate was required than under strictly anaerobic storage for >90 days. An effect of forage species can also not be ruled out as the chemical additive NHS, applied at the same rate of 2 L $t^{-1}$ as in our trial, improved the ASTA over that of untreated grass-clover and early-cut rye silage [18,57]. Moreover, as recently shown by da Silva et al. [64] in maize silage, the chemical additives used in our study may also likely have altered the qualitative composition of the yeast flora before the silo opening by causing a shift in the abundance from lactate assimilators (e.g., *Pichia kudriavzevii*) in untreated silage to non-lactate utilising species (e.g., *Candida humilis*) in treated silage, which usually do not show the capacity to assimilate lactic acid, or only grow slowly on this carbon source. Due to the high WSC concentration in NHS and BSP-treated silage upon silo opening, there was sufficient metabolizable substrate available for yeast growth by respiration during air ingress periods during storage and upon subsequent air exposure after silo opening, and yeasts grow faster on sugar than on lactic acid [66]. However, it remains to be elucidated why this, or physiologically similar, species did not ferment sugar to ethanol in the early stages of fermentation before first air ingress on day 28. Obviously, both chemical additives showed their potential to suppress the activity of all fermenting yeast species as reflected by the very low ethanol concentrations. This ethanol-reducing effect was observed for the

additive BSP (2 L t$^{-1}$) in maize silage already during the initial stages of fermentation from day 7 of storage and persisted until the silo opening after 142 days [34].

Another reason for the poor ASTA may have been the activity of acetic acid bacteria (AAB), which metabolise ethanol to form acetic acid under aerobic conditions and were shown to play a role in the aerobic deterioration of maize silage [67], but no data were available for whole-crop cereal silage. Along with air stress resulting in a higher relative abundance of Acetobacteraceae within the bacterial community of maize silage [64], several chemical silage additives may selectively inhibit yeasts, thereby creating conditions for AAB to cause aerobic spoilage [64,68]. The potential role AAB may play in the aerobic deterioration of whole-crop cereal silage warrants further attention in future ensiling experiments.

### 4.3. Formation of Volatile Organic Compounds

Ethyl lactate and ethyl acetate were detected in WCR silage, thereby further substantiating data from whole-crop oats [13], maize and sugarcane silage [33–35]. Higher concentrations of ethyl lactate than ethyl acetate were detected in our study, which agrees with results from whole-crop oats [13], sugarcane [35], and maize silage [30,32,33]. However, in other studies, more ethyl acetate was formed than ethyl lactate [34]. The discrepancy between studies regarding the relative abundance of individual ethyl esters may be explained by different underlying reaction pathways. As suggested by Weiss et al. [34], ethyl lactate is exclusively produced by a chemical reaction of ethanol and lactic acid, whereas ethyl acetate may only be partially chemically synthesised [30,34]. A certain, yet not quantified or quantifiable fraction, may originate from metabolic pathways of certain ester-producing yeasts [69,70], of which assimilators (e.g., *Pichia*, *Issatchenkia*) and non-assimilators (e.g., *Saccharomyces*) of lactic acid have been found in silages [71,72]. However, in light of generally low ethyl acetate concentrations, our data strongly suggest that ethyl ester-producing yeasts played no major role in the production of this ethyl ester and that ethanol was the main driver of ethyl acetate accumulation. This explanation is supported by the fact that lower ethanol levels have resulted in lower ethyl ester concentrations in numerous studies [13,30,31,73].

As shown by Hafner et al. [31,73] in maize silage and by Weiss [30] in various silage types including grass and sorghum silage, the use of homofermentative LAB in WCR also did not reduce ethyl ester concentration due to the lack of effect on ethanol formation. However, the strong ester-reducing effect of certain chemical ingredients, which has been shown for a range of forages [30–32,34,73], was confirmed by our WCR results. Thus, the application of this additive type, including, sorbic, benzoic and propionic acids and their salts, applied alone or in combination, can be considered a successful silage management tool to restrict VOC production. Our observations on the effects of *Lactobacillus buchneri*-containing inoculants agree with those presented by Weiss [30] from one sorghum study but contradict reports by others [13,31,73] showing no effect, or a stimulation of ester formation by this additive type. Obviously, only when the use of *Lactobacillus buchneri* led to a reduction in the concentration of ethanol, a lower silage ester content may be detected. In order to enlarge the body of evidence, further research should be directed at the potential of *Lactobacillus buchneri*-containing additives to modify the VOC production pattern.

Although the propyl ester of acetic acid has been detected in whole-crop oats [13] and maize silage [30,74], we did not find this compound in WCR silage, most probably due to the absence of the required reactant n-propanol.

### 4.4. Correlations between Silage Traits

The results of the regression analysis on anaerobic DM losses in WCR silage, which were mainly caused by the production of ethanol, are in line with those by others on whole-crop maize silage [64] and sugarcane silage [75], although the type of relationship may differ depending on the data set. We found a quadratic regression model as opposed to the linear relationships described Weiss et al. [34] in maize silage ($R^2 = 0.70$, $p < 0.001$) and by Rabelo et al. [74] in sugarcane silage ($R^2$ not given, $p < 0.01$).

Although ethanol formation by anaerobic sugar metabolism by yeasts accounts for the highest DM loss (59%) and the lowest DM recovery (51%), respectively, of all major fermentation pathways in silage [15], the activity of other ethanol-forming microorganisms will likely affect the type and the power of the relationship between DM losses and ethanol concentration determined in different studies.

The negative relationship between the yeast count and ASTA has been consistently demonstrated in meta-analyses in silages from grass and small grain silage [21], whole-crop maize [54] and a study using whole-crop rye before ear emergence [53], highlighting the prominent role of this microbial population to initiate the aerobic deterioration of silage [71]. However, the power of the relationship may vary greatly, especially when other microorganisms, e.g., acetic acid bacteria, are present in the silage and find favourable environmental conditions in which to thrive [67].

Acetic acid, which is the only desired short-chain fatty acid in silage having antifungal properties, as opposed to butyric, valeric and caproic acids from clostridia metabolism [15], was confirmed to have a significant effect on yeast count and, thus, ASTA. In agreement with Kleinschmit and Kung [21], a stimulation of acetic acid production reduced yeast count ($R^2 = 0.66$, $p < 0.01$) and, in turn increased ASTA ($R^2 = 0.95$, $p < 0.001$) [76], although the types of the relationships were linear. We detected quasilinear relationships, as were previously described by Auerbach et al. [53] in whole-crop rye silage harvested before ear emergence. Differences in the best-fit regression types may be attributed to forage species and the number and the types of additives used. In maize trials [21,76], only *Lactobacillus buchneri*-containing additives, alone or in combination with homofermentative LAB species were used, whereas Auerbach et al. [53] studied inoculants containing *Lactobacillus buchneri* and also one only composed of homofermentative strains, with the range being further extended by chemical additives in our WCR study. Obviously, the production of acetic acid, and that of other fermentation end-products, is affected by different additives according to their specific mode of action, which will also have an effect on the type and power of the regression.

As previously shown for other silage types [32,34,36], we also determined strong positive relationships between the concentrations of ethanol and ethyl esters in WCR silage, when single-point regressions (one storage length) were evaluated. However, as pointed out by Weiss et al. [34], the coefficient of determination may be much weaker (ethyl acetate: $R^2 = 0.35$, $p < 0.001$; ethyl lactate: $R^2 = 0.65$, $p < 0.001$) when ethyl ester concentrations were monitored several times during the course of the fermentation of maize silage over up to 142 days due to different accumulation pattern of ethanol and ethyl esters. Although ethanol and ethyl lactate remained at a similar level after their peaks, ethyl acetate declined over time. This observation was explained by the authors by different production pathways (chemical and/or biochemical) and by the higher vapour pressure of ethyl acetate than ethyl lactate, leading to losses of this compound with fermentation gas, escaping from the silo. Obviously, more studies are needed to study the relationships between ethanol and ethyl esters in whole-crop cereal silage, including rye.

An attempt was made to further validate the ethyl ester prediction model in silage developed by Weiss et al. [30]. This model is based on data from a total of 1148 silages from grass, whole-crop maize, whole-crop wheat, sorghum and high-moisture corn. They showed that each incremental increase in ethanol concentration (x) by 5 g kg$^{-1}$ DM resulted in an increase in the total ethyl ester content (y) by 114 mg kg$^{-1}$ DM ($R^2 = 0.76$). As demonstrated in Figure 4, the measured concentration of total ethyl esters in WCR silage and that predicted by the model showed a strong linear relationship ($R^2 = 0.93$, $p < 0.001$). Using this approach offers the advantage that no additional labour and specialised equipment is required, as ethanol is a routinely measured silage quality trait. However, the performance and robustness of this model requires validation on data from additional trials covering a range of silage types, including WCR silage.

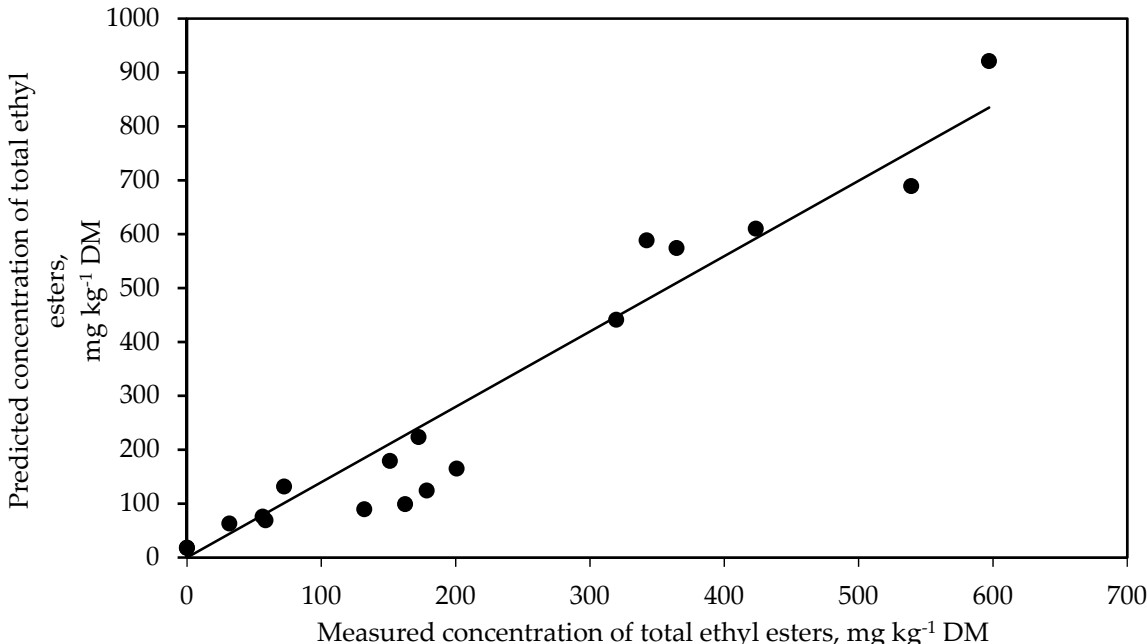

**Figure 4.** Relationship between the measured concentration of total ethyl esters (sum of ethyl lactate and ethyl acetate) and the predicted concentration of total ethyl esters (sum of ethyl lactate and ethyl acetate) based on Weiss et al. [30] (y = 1.40x, $R^2$ = 0.93, RMSE = 71.18, *p* < 0.001, *n* = 18) in whole-crop rye silage treated with different additives and stored for 53 days.

## 5. Conclusions

High losses of DM during the fermentation of WCR silage were caused by excessive ethanol production, accompanied by high concentrations of ethyl esters and rapid aerobic deterioration after silo opening. With the exception of homofermentative LAB, the use of all additives can be encouraged to reduce DM losses, improve ASTA and restrict ester formation, with *Lactobacillus buchneri*-containing additives best suitable to enhance ASTA and chemical additives showing the highest potential to minimize the formation of climate-relevant volatile organic compounds. Strong linear relationships existed between the concentrations of ethanol and those of ethyl acetate and ethyl lactate, respectively. Further studies are warranted to compare the influence of different additive types and compositions in terms of the consistency and magnitude of the effect in WCR silage, and to validate the ethanol-based model employed to predict ethyl ester formation in various silage types.

**Author Contributions:** Conceptualization, H.A. and P.T.; methodology, H.A. and K.W.; software, B.K.; validation H.A., K.W. and P.T.; formal analysis, B.K.; investigation, H.A. and K.W.; resources, P.T., H.A., K.W. and B.K.; data curation, H.A. and P.T.; writing—original draft preparation, H.A.; writing—review and editing, H.A., K.W. and B.K.; visualization, H.A.; supervision, K.W. and P.T.; project administration, H.A. and P.T.; funding acquisition, H.A. and P.T. All authors have read and agreed to the published version of the manuscript.

**Funding:** This research was primarily funded by the University Nürtingen-Geislingen, Germany, and the International Silage Consultancy (ISC), Germany. Additional minor funding was provided by the KONSIL Europe GmbH, Germany.

**Acknowledgments:** We would like to thank U. Mertin, Agrargenossenschaft Trebbin, Klein Schulzendorf, Germany, for kindly providing the forage used in this study. The support of W. Richardt, LKS mbH, Lichtenwalde, Germany, and his team is greatly acknowledged.

**Conflicts of Interest:** Horst Auerbach and Peter Theobald are holding shares in the company KONSIL Europe GmbH, which provided the tested additives.

## Abbreviations

ASTA, aerobic stability; EA, ethyl acetate; EL, ethyl lactate; DM, dry matter; WCR, whole-crop rye; LAB, lactic acid bacteria; VOCs, volatile organic compounds, WSCs, water-soluble carbohydrates

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
