# Peer review of "Effects of Various Additives on Fermentation, Aerobic Stability and Volatile Organic Compounds in Whole-Crop Rye Silage"

_agronomy, doi:10.3390/agronomy10121873_

Round 1
Reviewer 1 Report
My main concern about this manuscript is whether the treatment replicates are pseudo replicates and not true (statistical) replicates. Please see me comment on this below. Nevertheless, the manuscript has a nice flow and results are well presented. However, the discussion needs linguistic improvement as most sentences are long, making it difficult to follow. My specific comments are:
Line 79-80: the sentence is redundant, please delete it.
Line 110: please specify how you made the three replicates for each treatment. Did you apply each additive on a pile and then divide the pile into three silos? In this case, you have pseudo replicate.
Line 149: remove this section from Materials and Methods and incorporate it into the Discussion. In addition, elaborate about number of samples, type of forage, etc. used to derive this equation so that readers do not need to read the reference paper.
Line 236 &7: remove space and “,” after parentheses
Figure 3: needs legend to clarify which symbol represents which variable
Line 244-46: Move it to Discussion (together with Fig 4)
Line 253: delete “count”
Line 263: you have losses less than 5.2%. Rewrite the sentence.
Line 263-66: sentence is not clear, please rewrite
Line 287-88: remove “and strains, respectively”
Line 294-297: rewrite the sentence.
Line 316-318: long sentence. Please rewrite
Author Response
Reviewer 1
My main concern about this manuscript is whether the treatment replicates are pseudo replicates and not true (statistical) replicates. Please see me comment on this below. Nevertheless, the manuscript has a nice flow and results are well presented. However, the discussion needs linguistic improvement as most sentences are long, making it difficult to follow.
AU: We have made numerous linguistic changes by shortening sentences, where possible and appropriate.
My specific comments are:
Line 79-80: the sentence is redundant, please delete it.
AU: Sentence deleted.
Line 110: please specify how you made the three replicates for each treatment. Did you apply each additive on a pile and then divide the pile into three silos? In this case, you have pseudo replicate.
AU: Method of additive application specified and explained.
Line 149: remove this section from Materials and Methods and incorporate it into the Discussion. In addition, elaborate about number of samples, type of forage, etc. used to derive this equation so that readers do not need to read the reference paper.
Line 236 &7: remove space and “,” after parentheses
AU: Done.
Figure 3: needs legend to clarify which symbol represents which variable
AU: Symbols included.
Line 244-46: Move it to Discussion (together with Fig 4)
AU: Moved to Discussion.
Line 253: delete “count”
AU: Deleted.
Line 263: you have losses less than 5.2%. Rewrite the sentence.
AU: rephrased.
Line 263-66: sentence is not clear, please rewrite
AU: rephrased.
Line 287-88: remove “and strains, respectively”
AU: Deleted.
Line 294-297: rewrite the sentence.
AU: Rewritten.
Line 316-318: long sentence. Please rewrite
AU: Rephrased.
Reviewer 2 Report
The paper is well written. Material and methods sound and clearly described. The results as expected and not new. The discussion profound and sound.
General comments:
The conclusions regarding the comparison between silage additives and in particular the correlation between silage traits are valid and sound. However, to draw the conclusion that this is valid for whole crop rye (Secale cereale L.) in general and compared to other crops is not valid since the sample used in this experiment is just one crop, from one field harvested a single year. In order to make statement for whole crop rye silage as such a wider spectrum of samples whole crop rye silages is needed. E.g. the epiphytic field flora, particularly the yeasts, could have very big influence on the result. This is mentioned but could be more clearly stressed.
Specific comments:
Line Comment
44 Rye silage is not completely un-published. There are even some publications with chemical additive to whole crop silage (Tetlow and Mason). See:
forage rye:
- Herrmann, Heiermann and Idler. 2011. Effects of ensiling, silage additives and storage period on methane formation of biogas crops. Bioresource Technology Volume 102: 8, pp5153-5161
- Lee, K.N. and Kim, D.A. 1997. Effects of wilting and additives on the fermentation chrateristics, quality and aerobic Sstability of rye silage. Journal of the Korean Society of Grassland and Forage Science. 17:187-198.
- Kim, J.G., Kim, D.A., Jung, E.S., Kang, W.S., Ham, J.S. and Seo. S. 1999. Effect of maturity at harvest and inoculants on the quality of round baled rye silage. Journal of the Korean Society of Grassland and Forage Science. 19:347-354.
- Choi et al. 2015. Effect of Addition of Lactic Acid Bacteria on Fermentation Quality of Rye Silage. Journal of The Korean Society of Grassland and Forage Science Volume 35 Issue 4 Pages.277-282
whole crop rye:
- R.M. Tetlow and V.C. Mason. 1987. Treatment of whole-crop cereals with alkali. 1. The influence of crop maturity, moisture content and treatment with sodium hydroxide on the chemical composition, preservation and in vitro digestibility of ensiled rye, barley and wheat Anim. Feed. Sci. Technol., 18 (1987), pp. 257-269
Line Comment
93-94 Exceptional low CP content. 4.4%. It is never under 10% in German feed tables.
120 Ref 42 difficult to get.
158-159 The model used makes it only relevant to compare the impact of the additives on a single batch of WCR. Not as for whole crop rye silage generally and there is no comparison with other crops.
186 No propionic acid detected. But propionic acid was added by additive BSP. Comment?
248-249 The signs (circles, squares) are not present in the table text.
328 The discussion of aerobic stability goes in detail. However, it should be kept in mind that it is only one sample from one field from one year with its specific epiphytic field flora of yeasts. It could be more clearly stressed.
422 The ref 13 is not Auerbach. Should it be ref 53?
426 The ref 13 is not Auerbach. Should it be ref 53?
Author Response
Reviewer 2
The paper is well written. Material and methods sound and clearly described. The results as expected and not new. The discussion profound and sound.
AU: We do not quite understand the doubts of the reviewer expressed regarding the novelty of the results. To our knowledge, data on the effects of a range of additives of whole-crop rye are still lacking (see below) and no study has focused on VOC formation in this substrate.
General comments:
The conclusions regarding the comparison between silage additives and in particular the correlation between silage traits are valid and sound. However, to draw the conclusion that this is valid for whole crop rye (Secale cereale L.) in general and compared to other crops is not valid since the sample used in this experiment is just one crop, from one field harvested a single year. In order to make statement for whole crop rye silage as such a wider spectrum of samples whole crop rye silages is needed. E.g. the epiphytic field flora, particularly the yeasts, could have very big influence on the result. This is mentioned but could be more clearly stressed.
AU: We see the point raised by the reviewer and want to stress that we already stated in the conclusions that more studies are needed to enlarge the body of knowledge on WCR silage. We simply need them to evaluate the consistency of the effects we have seen, but we believe that our paper forms a good basis and our data are worth publishing.
We also acknowledge the role the epiphytic microflora may play regarding fermentation pattern and aerobic stability. However, as shown by Mogodiniyai Kasmaei et al. (2013), doi: 10.3168/jds.2013-6858, models based on available chemical and microbial crop variables had limited explanatory value to predict the outcome of the fermentation process, and preliminary results by Mogodiniyai Kasmaei et al. (2016), doi: 10.1111/gfs.12238, suggested that fermentation quality is mainly affected by forage source, whereas the aerobic stability is affected by both forage and field microbiota. We also want to highlight the effect of storage conditions (fermentation length and air ingress during fermentation) which will affect silage traits even if the same batch of crop is used.
Specific comments:
Line Comment
44 Rye silage is not completely un-published. There are even some publications with chemical additive to whole crop silage (Tetlow and Mason).
AU: We would like to thank the reviewer for providing the references the authors have been fully aware of. However, there were reasons for not citing them (see below).
forage rye:
- Herrmann, Heiermann and Idler. 2011. Effects of ensiling, silage additives and storage period on methane formation of biogas crops. Bioresource Technology Volume 102: 8, pp5153-5161
AU: This paper used forage rye and whole-crop triticale. Due to differences in chemical composition, we have cited some forage rye studies but not this one in order to restrict number of references. Also, whole-crop triticale is a different forage compared to whole-crop rye. So, when deemed appropriate, we have cited studies on barley and wheat instead.
- Lee, K.N. and Kim, D.A. 1997. Effects of wilting and additives on the fermentation chrateristics, quality and aerobic Sstability of rye silage. Journal of the Korean Society of Grassland and Forage Science. 17:187-198.
AU: We did not cite this paper because it was written in Korean, and we do not understand this language. Based on Tables and the Summary (in English) the paper was on rye harvested at heading which is different to whole-crop rye.
- Kim, J.G., Kim, D.A., Jung, E.S., Kang, W.S., Ham, J.S. and Seo. S. 1999. Effect of maturity at harvest and inoculants on the quality of round baled rye silage. Journal of the Korean Society of Grassland and Forage Science. 19:347-354.
AU: Also in (mainly) Korean. The latest maturity stage tested was flowering stage, so much earlier than the whole-crop rye we used.
- Choi et al. 2015. Effect of Addition of Lactic Acid Bacteria on Fermentation Quality of Rye Silage. Journal of The Korean Society of Grassland and Forage Science Volume 35 Issue 4 Pages.277-282
AU: Also (mainly) in Korean. Study used forage rye.
whole crop rye:
- R.M. Tetlow and V.C. Mason. 1987. Treatment of whole-crop cereals with alkali. 1. The influence of crop maturity, moisture content and treatment with sodium hydroxide on the chemical composition, preservation and in vitro digestibility of ensiled rye, barley and wheat Anim. Feed. Sci. Technol., 18 (1987), pp. 257-269
AU: This study was included in the paper (replaced former reference no 7 by Orosz et al, 2018) because it supports the changes in nutrient composition as affected by stage of maturity and provides data on fermentation pattern of whole-crop rye silage. However, this paper cannot be cited with regard to additive effects. The reason is that the use of NaOH (or ammonia) preserves the crop by alkalinization which is completely different compared to fermentation aids, including inoculants or salts of organic/inorganic acids.
Line Comment
93-94 Exceptional low CP content. 4.4%. It is never under 10% in German feed tables.
AU: We see the validity of this point raised but would like to stress that CP content can largely vary as affected by a range of factors, including N fertilization and variety. The variety used in our study is mainly used for forage rye production. This may contribute to explaining the low CP value. However, as outlined above (Mogodiniyai Kasmaei et al., 2013, doi: 10.3168/jds.2013-6858) prediction of the outcome of the fermentation process based on chemical composition of the forage used has serious constraints. Furthermore, we have to be careful in using the feed tables as they only give an average value across a number of studies and do not report the min/max values, but only SD. Thus, it may well be possible that the data set in the German feed tables contains whole-crop rye with exceptionally low CP content.
120 Ref 42 difficult to get.
AU: The equations used are approved by the German Gesellschaft für Ernährungsphysiologie (GfE). For questions and receipt of a copy please contact Dr. Detlef Kampf, Tel.: 069/24788-320, Mail: d.kampf@dlg.org.
158-159 The model used makes it only relevant to compare the impact of the additives on a single batch of WCR. Not as for whole crop rye silage generally and there is no comparison with other crops.
AU: The statistical model was a 1way-ANOVA with silage additive as the only factor. This model is suitable to detect the effects of additives on the silage traits tested.
186 No propionic acid detected. But propionic acid was added by additive BSP. Comment?
AU: Considering the used application rate of additive BSP of 1.5 L/t and the concentration of ammonium propionate of 57 g/L (=46.3 g/L propionate) in the product, theoretically 70 g/t of propionic acid were added to the forage, which amounts to 0.007%. This is below the detection limit of 0.01% of fresh matter, thereby rendering this compound undetectable by the employed method.
248-249 The signs (circles, squares) are not present in the table text.
AU: Symbols added.
328 The discussion of aerobic stability goes in detail. However, it should be kept in mind that it is only one sample from one field from one year with its specific epiphytic field flora of yeasts. It could be more clearly stressed.
AU: We do understand that drawing general conclusions based on just one trial is not justified. However, we believe that we did not generalize. Aerobic stability as one of the key silage quality traits has been discussed thoroughly because of the results, which have not been expected based on what has been published in the literature, especially when the effects of L. buchneri and chemical additives are compared. Indeed, we believe that our findings are very interesting and justify discussion of possible and likely reasons for our observations.
422 The ref 13 is not Auerbach. Should it be ref 53?
AU: reference number changed to 53.
426 The ref 13 is not Auerbach. Should it be ref 53?
AU: reference number changed to 53.
Round 2
Reviewer 1 Report
I would like to congratulate authors for the improvement. Nevertheless, there are still some parts that need improvement. Please see my comments below.
Line 32: add “developed by Wiess (30)” after ethanol content.
Line 62: remove maize as maize is a cereal.
Line 100-110: you have added 10 ml microbial solution/tap water per kg forage for your biological/control treatments but have added 2 and 1.5 ml chemical solution per kg forage for the chemical treatments. I believe this is an error in the experimental setup.
Line 244: typo “extisted”
Line 285: typo “und”
Line 350-359: remove this section. This is because first aerobic stability was improved in chemical treated silages compared to the control in your study. And second, if the chemical additives does not change yeast microflora from non-lactate assimilating to lactate assimilating, what the point of this argument is.
Line 371: change “whole-crop cereals silage” to “whole-crop cereal silages”
Line 379: “but…………..sugar cane silage” is unclear. Please rewrite the whole sentence.
Line 435-437: the sentence “In addition……….WCR study.” is unclear. Please rewrite or delete.
Line 443-444 : the sentence “However………..much weaker.” is hanging loose. Please blend the sentence with the succeeding sentence.
Line 453-460: this section is hanging loose. I suggest to start the section as the following or similar: an attempt was made to further validate ethyl ester prediction model in silage developed by Weiss (30). The model is based on data from a total of ………………………… As shown in Figure 4………
Figure 4: include in the caption that the predicted concentration is based on Weiss (30).
Line 480-482: remove the sentence. Instead, add a sentence that there was a strong relation between ethanol and ethyl ester formation in your study and that the model of Weiss (30) based on ethanol concentration can be used to predict ethyl esters in silage.
Author Response
I would like to congratulate authors for the improvement. Nevertheless, there are still some parts that need improvement. Please see my comments below.
AU: We deeply acknowledge the referee´s efforts to help us improve the paper. Many thanks for all your valuable comments and suggestions.
Line 32: add “developed by Wiess (30)” after ethanol content.
AU: We would want to keep our original version of the abstract because we truly think that a reference has no place in an abstract, and not citing Weiss et al. will not make the abstract less clear. The model we used is explained in detail in the text. We do hope that the referee can agree upon our approach. If not, we would like to learn why he/she, specifically, suggested the inclusion of the reference.
Line 62: remove maize as maize is a cereal.
AU: We do see where the referee is coming from but would like to keep our original wording. In the silage community, when one uses the term whole-crop cereal silage then everybody will know that those made from oats, barley, rye, wheat are referred to. Silage made from whole-plant maize is not considered to be a WCC. In fact, botanically, even grass would be a cereal as all of the afrorementioned plants are grasses botanically (family Gramineae).
Line 100-110: you have added 10 ml microbial solution/tap water per kg forage for your biological/control treatments but have added 2 and 1.5 ml chemical solution per kg forage for the chemical treatments. I believe this is an error in the experimental setup.
AU: This section was re-phrased. However, we did not make a mistake in experimental set-up. Inoculants were suspended in 10 ml water/kg forage, whereas all chemicals were diluted with water to give 10 ml/kg forage.
Line 244: typo “extisted”
AU: Changed.
Line 285: typo “und”
AU: Changed.
Line 350-359: remove this section. This is because first aerobic stability was improved in chemical treated silages compared to the control in your study. And second, if the chemical additives does not change yeast microflora from non-lactate assimilating to lactate assimilating, what the point of this argument is.
AU: We have shortened and reworded this section. Unfortunately, there was a mistake in the text caused by the prefix “non”, which may have been the reason that caused the confusion. Pichia a lactate-assimilators and not, as stated, non-assimilators. We are truly sorry for this error.
However, we still believe that, although a hypothesis, the potential changes in qualitative composition of the yeast flora may provide a valuable and plausible explanation for the lower than usual magnitude of the effect of chemical additives on ASTA. We clearly stated that chemicals improved ASTA but this effect was only weak (and weaker than usually found). There are several possible reasons for this finding which we all addressed. We hope that the referee finds the way this section is worded now acceptable.
Line 371: change “whole-crop cereals silage” to “whole-crop cereal silages”
AU: Changed.
Line 379: “but…………..sugar cane silage” is unclear. Please rewrite the whole sentence.
AU: Changed.
Line 435-437: the sentence “In addition……….WCR study.” is unclear. Please rewrite or delete.
AU: Reworded.
Line 443-444 : the sentence “However………..much weaker.” is hanging loose. Please blend the sentence with the succeeding sentence.
AU: Rephrased, sentences blended.
Line 453-460: this section is hanging loose. I suggest to start the section as the following or similar: an attempt was made to further validate ethyl ester prediction model in silage developed by Weiss (30). The model is based on data from a total of ………………………… As shown in Figure 4………
AU: Rephrased based on the referee´s suggestion.
Figure 4: include in the caption that the predicted concentration is based on Weiss (30).
AU: Reference included.
Line 480-482: remove the sentence. Instead, add a sentence that there was a strong relation between ethanol and ethyl ester formation in your study and that the model of Weiss (30) based on ethanol concentration can be used to predict ethyl esters in silage.
AU: We added a sentence on the relationship between ethanol and ethyl esters, which are, indeed, an important finding of our study. However, we are hesitant to use the wording suggested by the referee to say that the model can be used to predict ethyl ester concentration. This would be too general because of a lack of other studies using this model. We rather prefer to amend the last sentence of the conclusions by saying that more studies on various silage types are needed to validate the ethanol-based model. We hope that the reviewer can agree now on our conclusions.